# Frontal Sinus Trephination and Repeated Irrigation in a Cat with Chronic Rhinosinusitis: A Case Report

**DOI:** 10.3390/ani15101382

**Published:** 2025-05-10

**Authors:** Hyomi Jang, Hyojun Kwon, Sunyoung Kim, Jiheui Sohn, Jong-in Kim, Dong-In Jung

**Affiliations:** 1VIP Animal Medical Center (Cheongdam), Seoul 06068, Republic of Korea; hm100su@gmail.com (H.J.); hyojun1220@naver.com (H.K.); duddl430@naver.com (S.K.); wlgml924@naver.com (J.S.); jjongin88@vipah.co.kr (J.-i.K.); 2College of Veterinary Medicine, Gyeongsang National University, Jinju 52828, Republic of Korea

**Keywords:** cat, chronic rhinosinusitis, frontal sinus, rhinitis, trephination

## Abstract

Chronic rhinosinusitis is a challenging condition in cats, and is often resistant to traditional therapies. This report describes a case in which a cat underwent frontal sinus surgery and repeated flushing for the treatment of chronic rhinosinusitis, leading to a significant improvement in symptoms for two years. This case highlights that surgery may be an important treatment option for chronic rhinosinusitis when typical treatments are ineffective, thus improving the quality-of-life of cats.

## 1. Introduction

Feline chronic rhinitis (FCR) is defined as inflammation within the nasal cavity persisting for more than four weeks or recurring intermittently [1,2]. When inflammation extends to the frontal sinuses, it is referred to as feline chronic rhinosinusitis (FCRS). The most common clinical manifestation of FCR is chronic, typically bilateral mucopurulent nasal discharge, often accompanied by sneezing and stertors. Facial asymmetry and conjunctivitis have also been observed [1,2]. Although FCR commonly affects young-to middle-aged cats, it can occur at any age, and tends to be progressive in nature [2,3].

However, its pathogenesis remains unclear. A recent hypothesis proposed that primary viral infections, such as those caused by feline herpesvirus-1 (FHV-1) or feline calicivirus, compromise the mucosal epithelium and turbinate bones, thereby increasing susceptibility to secondary bacterial infections, which subsequently leads to anatomical alterations of the upper respiratory tract and impaired immune function [2,4,5,6,7]. Therefore, the diagnosis currently focuses on excluding underlying conditions that might cause chronic upper respiratory symptoms, such as primary viral or bacterial infections, fungal infections, nasal parasites or tumors, congenital anomalies, dental diseases, nasal foreign bodies, nasopharyngeal polyps, allergic rhinitis, and trauma [1,2,8,9]. Additionally, diagnosis aims to assess the severity of lesions and the extent of anatomical changes [1,2,8,9].

Currently, no definitive cure exists for FCR. Medical management, including antibiotics, antifungal agents, antivirals, and anti-inflammatory drugs, remain the most commonly employed therapeutic approach for alleviating or managing symptoms [2,3,10]. However, recurrent clinical signs and long-term medical management can significantly impair the patient’s quality-of-life [2,11]. Surgical intervention may be indicated in patients with persistent or worsening symptoms. Techniques such as turbinectomy, frontal sinus ablation, and sinus trephination facilitate endoscopic access to the frontal sinus and nasal cavity, promote effective drainage and flushing to remove exudate and necrotic debris, and allow for sample collection, which is necessary for culture and histopathological evaluation [7,10,12,13].

This case report describes the diagnosis, surgical management, and outcomes of a cat with chronic and progressively worsening upper respiratory symptoms despite several years of medical therapy.

## 2. Case Description

A 2-year-old neutered female domestic shorthair cat presented with chronic upper respiratory symptoms. The patient was rescued from the streets a year earlier and was diagnosed with feline herpesvirus-1 (FHV-1). The diagnosis of feline herpesvirus-1 (FHV-1) was based on the medical history provided by the owner, who reported that the cat had been diagnosed with FHV-1 and treated with famciclovir at a previous veterinary clinic. However, the specific diagnostic method used (e.g., PCR or viral culture) could not be confirmed. Despite multiple treatment attempts over time, including the use of antibiotics, famciclovir, and glucocorticoids either separately or in combination, the cat continued to exhibit persistent mucopurulent nasal discharge and sneezing that were refractory to therapy.

Physical examination revealed a normal body temperature. The complete blood count (CBC) and serum biochemistry profiles were normal (Table 1). Head computed tomography (CT) revealed fluid-filled, non-contrast-enhancing material within the nasal cavities, nasopharynx, nasal turbinates, sphenoid sinuses, and frontal sinuses, with no signs of turbinate bone loss (Figure 1).

Nasal flushing was performed while the patient was intubated with a tracheal tube. After disinfecting the nasal region, the patient was positioned in sternal recumbency with the head angled downward. Sterile saline was forcefully flushed into both nostrils using a 10 mL syringe. The resulting lavage fluid and exudate drained through the nasal and oral cavities. Purulent nasal discharge exiting from the nostrils was collected using a sterile swab and transferred into a PCR-specific transport medium.

PCR analysis of the discharged exudate revealed *Streptococcus* spp. and *Pseudomonas aeruginosa*. Bacterial and fungal cultures were negative. Cytological examination revealed numerous neutrophils, epithelial cells, and a few cocci in the sample. Medical therapy included marbofloxacin (2 mg/kg PO q24h), doxycycline (5 mg/kg PO q12h), azithromycin (5 mg/kg PO q24h, tapered to q72h after five days), bromhexine (1 mg/kg PO q12h), and omeprazole (1 mg/kg PO q12h). Prednisolone was administered at an anti-inflammatory dose (0.5 mg/kg PO q12h) and gradually tapered to 0.3 mg/kg PO q12h over approximately one month. Additionally, at-home nebulization with sterile saline was recommended once or twice daily.

However, no significant clinical improvement was observed after 35 days of medical treatment, prompting repeat CT imaging and nasal flushing under anesthesia.

Follow-up CT revealed significant atrophic changes in the right nasal turbinates and localized osteolytic changes in the palatine bone, forming the sphenoid sinuses (Figure 2). Similarly to previous findings, a fluid-attenuating material was observed in the nasal cavities and frontal sinuses. Antifungal therapy (terbinafine, 30 mg/kg PO q12h) was added, but the clinical signs did not improve after one month, leading to the decision to perform frontal sinus trephination.

Frontal sinus trephination was performed using a 2.0 mm pin drill, followed by curettage to enlarge the sinus opening. Two openings were created, revealing a highly viscous, purulent exudate that required removal using sterile cotton swabs (Figure 3). Endoscopic examination after removal revealed mucosal edema and petechiae (Figure 4A). Biopsy was thus avoided due to easy bleeding from the fragile mucosa. Extensive flushing with sterile saline was performed until no further exudate emerged (Figure 4B). An 8 Fr red rubber catheter was inserted into the frontal sinus and fixed to the skin (Figure 4C). PCR testing of the exudate revealed *Aspergillus fumigatus* and *Mycoplasma felis*, whereas bacterial and fungal cultures were negative.

During the seven days of hospitalization post-surgery, two additional nasal sinus flushings were performed at 3–4 day intervals. The cat was treated with terbinafine (30 mg/kg PO, q24h), fluconazole (1.25 mg/kg PO, q12h), clindamycin (11 mg/kg PO q12h), doxycycline (5 mg/kg PO q12h), and azithromycin (7 mg/kg PO q72h). CT revealed a significant reduction in sinus inflammation and exudate accumulation (Figure 5A). The rubber tube was removed, and skin closure was performed at discharge. After discharge, frontal sinus irrigation was conducted through the remaining skin openings using 0.1% povidone–iodine and sterile saline one week later, and repeated three weeks thereafter. PCR analysis of the final flush was negative for *Aspergillus fumigatus* and *Mycoplasma felis*. No adverse effects other than mild facial swelling due to subcutaneous emphysema were observed during the repeated incision–flush–suture procedures (Figure 5B).

The mucopurulent nasal discharge gradually became serous over two months, allowing for gradual tapering and eventual cessation of the medications. Follow-up at two years post-procedure showed no significant clinical signs, aside from intermittent serous nasal discharge.

## 3. Discussion

The treatment of FCR or FCRS is challenging because of difficulties in identifying the primary cause and the rarity of complete resolution despite treatment. Frequent recurrence can reduce patient and caregiver treatment compliance, and potentially lead to inadequate disease management [2,7,10]. Poor management can result in complications such as otitis media, severe bone lysis, and central nervous system damage through erosion of the cribriform plate, ultimately necessitating euthanasia [2,11].

In the present case, the patient had a previous diagnosis of feline herpesvirus-1 (FHV-1), which was initially considered a potential underlying cause. However, the lack of clinical improvement despite long-term famciclovir administration, combined with two negative PCR results for FHV-1 and the marked clinical improvement achieved at our hospital without the use of antiviral therapy, suggest that FHV-1 was unlikely to have played a direct role in the ongoing clinical signs.

Initial PCR analysis of the nasal exudate was positive for *Streptococcus* spp. and *Pseudomonas aeruginosa*; however, cytological examination revealed only cocci, and neither organism was isolated from bacterial culture. Subsequent PCR of the frontal sinus exudate confirmed *Aspergillus fumigatus* and *Mycoplasma felis*; however, fungal and bacterial cultures remained negative.

Previous studies have shown that bacterial populations differ between healthy cats and those presenting with upper respiratory signs, with *Streptococcus*, *Mycoplasma* spp., *Pasteurella*, *Staphylococcus*, *Chlamydia*, and *Moraxella* frequently identified in symptomatic individuals [14,15,16]. Additionally, *Pseudomonas aeruginosa* and *Aspergillus fumigatus* have been implicated as pathogens in cases of severe FCR [17,18,19]. Notably, definitive pathogen identification using PCR and culture methods has inherent limitations.

First, in FCR, bacterial infections typically occur secondary to mucosal injury and immunosuppression. Moreover, some of the identified bacteria may also be present in healthy individuals, making it difficult to assess their clinical relevance [2,4,16]. Previous reports have described temporary improvement in FCR clinical signs following antibiotic therapy targeting suspected pathogens; however, clinical signs often recurred, requiring long-term management with immunomodulatory drugs [17]. In particular, although PCR testing is a highly sensitive diagnostic method for detecting pathogens, it is more likely to detect normal flora or secondarily colonizing bacteria rather than the primary pathogen in nasal samples. The presence or relative abundance of microorganisms identified via PCR does not necessarily indicate pathogenicity [18]. It is also important to note that the detection of microorganisms by PCR does not necessarily imply pathogenicity, as some organisms may represent normal flora or secondary colonizers. Therefore, PCR findings must be interpreted cautiously and in conjunction with clinical signs, imaging, and treatment response. Similarly, culture-based diagnostics may yield false-negative results due to prior antimicrobial therapy or fastidious growth requirements, underscoring the need for a multimodal diagnostic approach in FCR cases.

In this case, clinical symptoms worsened despite treatment and adjustment based on PCR results, suggesting that pathogen-targeted therapy alone may not be sufficient for clinical recovery in FCR.

In addition, due to the chronic nature of FCR, patients are often exposed to antibiotics or antifungal agents for prolonged periods, which may reduce the viability of pathogens in culture media. Moreover, the involvement of fastidious anaerobic bacteria cannot be excluded, as commonly used hospital transport systems and standard laboratory growth media may not provide optimal conditions for their proliferation [16]. In the present case, the inability to consistently isolate suspected pathogens through culture is presumed to be attributable to these factors.

Lastly, differences in pathogen detection based on sample location and diagnostic methods have been reported. Pathogens that are undetectable by PCR or culture are sometimes identified only by histopathology [16,19,20]. Similarly, this case exhibited varied pathogen identifications based on the sampling sites and diagnostic methods, highlighting the diagnostic limitations of FCR. A comprehensive evaluation integrating PCR, culture, and histopathology from multiple sites, along with clinical response assessment, could mitigate these limitations. However, anesthetic risks, financial constraints, and potential complications, such as bleeding, limit the routine extensive use of these diagnostics.

This case illustrates the potential benefits of surgical intervention for refractory FCRS, in which medical management alone has been unsuccessful. Persistent viscous exudate obstructing the sinuses and nasal passages responded inadequately to medical treatment, but improved significantly following physical removal and repeated irrigation. Similarly, a previous study [21] reported successful management of FCR using frontal sinus trephination, exudate removal, and placement of a sinus-to-nasal stent, resulting in sustained clinical resolution without recurrence of obstruction at 20 months of follow-up. In humans, sinonasal obstruction is often effectively resolved using endoscopic endonasal surgery, resulting in favorable outcomes [22,23,24]. However, veterinary reports have indicated residual mild symptoms or lesion recurrence, suggesting that treatment goals should focus on symptom alleviation and quality-of-life improvement by extending drug-free intervals [19,21].

Seventeen cats diagnosed with FCR or FCRS have been treated using frontal sinus obliteration with bone cement and radical turbinectomy [25]. Among them, two cats showed marked clinical improvement, twelve exhibited a 10–70% reduction in clinical signs, and three showed minimal or no response despite surgical intervention. Another reported approach involved ethmoid conchal curettage combined with autogenous fat graft sinus ablation, in which two of six cases demonstrated clinical improvement [13]. Although these techniques may offer more definitive resolution compared to temporary frontal sinus trephination, they are significantly more invasive, and may not result in consistent clinical benefits. Therefore, such procedures may be considered as alternative options in cases where symptoms persist or recur following initial sinus trephination and flushing.

A recent study reported that three cats diagnosed with infectious rhinosinusitis underwent rhinotomy and irrigation, followed by nasal packing with 5% povidone-iodine, which was replaced after 24 h and then completely removed 48 h later [26]. Subsequent medical management, including the administration of antibiotics and antifungal agents, resulted in complete resolution of the symptoms without adverse effects. Therefore, surgical intervention and nasal packing with 5% povidone-iodine may be an effective treatment option for infectious rhinosinusitis.

## 4. Conclusions

In this case, frontal sinus trephination, placement of a drainage tube, and repeated irrigation significantly improved the clinical signs in an FCRS patient who was unresponsive to medical therapy. Over an approximately 2-year follow-up period, the patient exhibited intermittent mild serous nasal discharge, but demonstrated overall clinical improvement. This report emphasizes surgical intervention as a critical therapeutic option for refractory FCR, highlighting the need for proactive treatment strategies when medical management is ineffective. Further studies are necessary to elucidate FCR pathophysiology and help establish standardized medical and surgical protocols, potentially enhancing treatment outcomes and improving patients’ quality-of-life.

## Figures and Tables

**Figure 1 animals-15-01382-f001:**
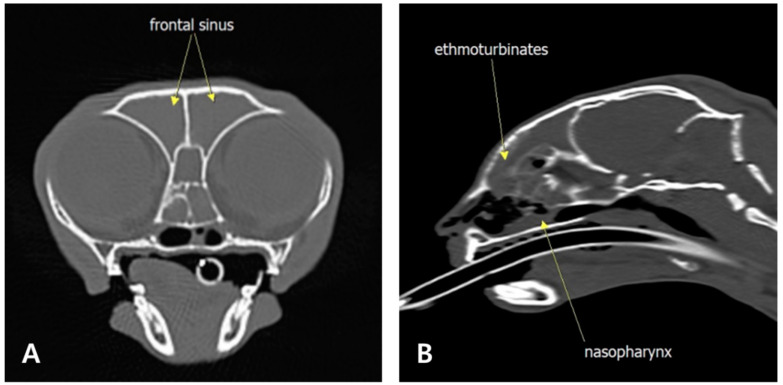
Computed tomography findings on day 1 (pre-contrast). Transverse plane (**A**) and sagittal plane (**B**) showing non-contrast-enhancing material filling the bilateral nasopharynx, nasal cavity, turbinates, ethmoturbinates, and frontal sinuses.

**Figure 2 animals-15-01382-f002:**
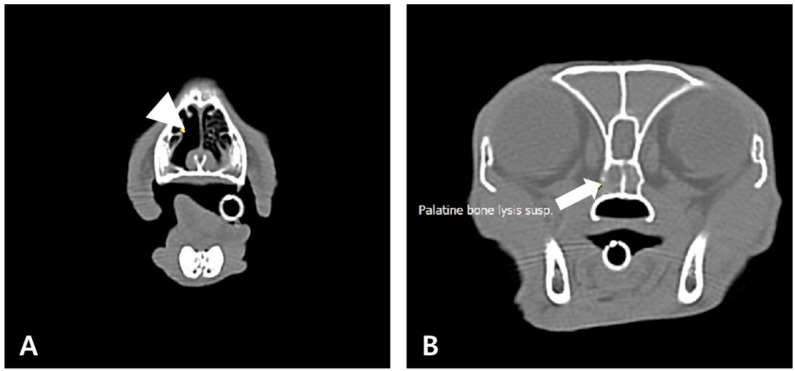
Computed tomography findings on day 35 (pre-contrast). Atrophic changes observed in the nasal turbinates ((**A**), arrowhead), demonstrating lysis of the palatine bone ((**B**), arrow). Fluid-attenuating material observed filling the nasal cavity and frontal sinus lumen, consistent with previous findings (**B**).

**Figure 3 animals-15-01382-f003:**
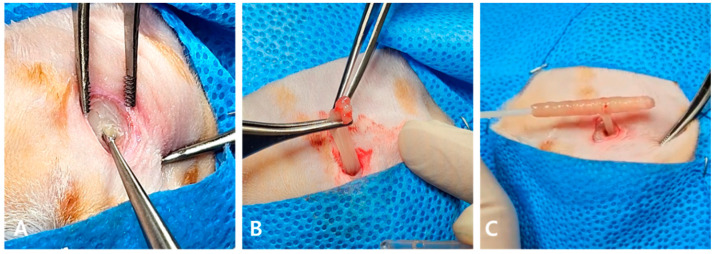
Frontal sinus trephination. Openings created in the frontal sinus (**A**). Frontal sinuses filled with highly viscous, solid purulent discharge (**B**), which was removed by rolling it out using cotton-tipped applicators (**C**).

**Figure 4 animals-15-01382-f004:**
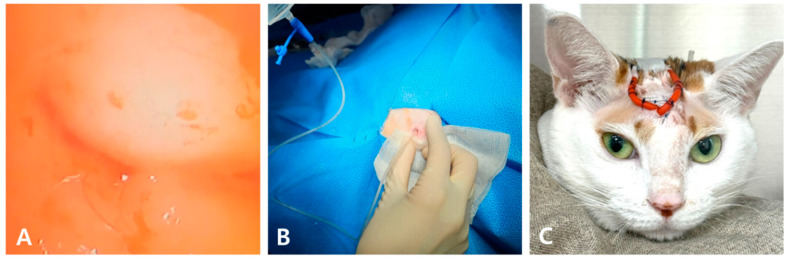
Severe mucosal edema and petechial hemorrhage were observed during rhinoscopy (**A**). Drainage with sterile saline, which was repeatedly conducted using a feeding tube (**B**). 8 Fr red rubber catheter placed in the frontal sinus and secured to the skin (**C**).

**Figure 5 animals-15-01382-f005:**
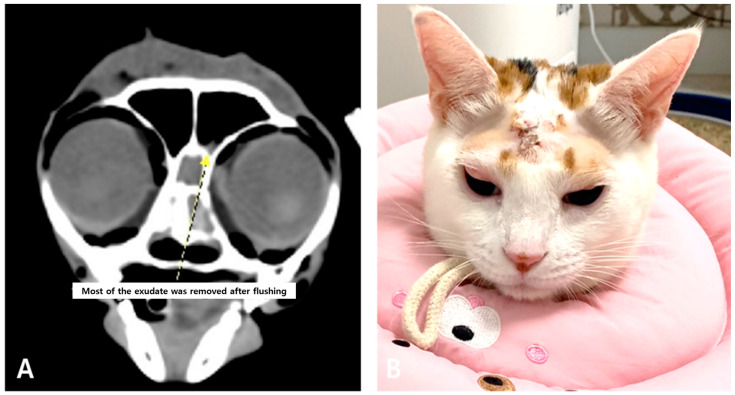
Pre-discharge computed tomography scans (pre-contrast) showing improvement in frontal sinus obstruction, with a small amount of residual lavage fluid (**A**). Facial swelling caused by subcutaneous emphysema (**B**).

**Table 1 animals-15-01382-t001:** Hematological results of the cat.

	Result	Reference Range
Hematocrit (%)	48.7	30.3–52.3
White Blood Cells (k/μL)	10.07	2.87–17.02
Neutrophils (k/μL)	4.5	2.30–10.29
Lymphocytes (k/μL)	4.82	0.92–6.88
Monocytes (k/μL)	0.22	0.05–0.67
Eosinophils (k/μL)	0.47	0.17–1.57
Platelet (k/μL)	262	151–600
Glucose (mg/dL)	104	74–159
BUN (mg/dL)	21	16–36
Creatinine (mg/dL)	1.3	0.8–2.4
ALT (U/L)	67	12–130
ALP (U/L)	28	14–111
GGT (U/L)	0	0–4
Total protein (g/dL)	8.2	5.7–8.9
Albumin (g/dL)	3.3	2.2–4.0
Globulin (g/dL)	4.9	2.8–5.1
Phosphorus (mg/dL)	4.7	3.1–7.5
Calcium (mg/dL)	10.1	7.8–11.3

BUN = Blood Urea Nitrogen, ALT = Alanine Aminotransferase, ALP = Alkaline Phosphatase, GGT = Gamma-Glutamyl Transferase.

## Data Availability

The original contributions of this study are presented. Further inquiries can be directed to the corresponding author.

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
