# Peer review of "Frontal Sinus Trephination and Repeated Irrigation in a Cat with Chronic Rhinosinusitis: A Case Report"

_animals, 2025, doi:10.3390/ani15101382_

Round 1
Reviewer 1 Report
Comments and Suggestions for Authors
The article addresses a clinically relevant issue (feline chronic rhinosinusitis - FCRS) with practical significance for veterinary clinicians.
Comments:
The rationale for prolonged glucocorticoid use, despite potential immunosuppressive side effects, was not discussed.
Limited discussion on why repeated bacterial cultures were negative despite positive PCR findings.
Limited discussion of alternative surgical approaches or comparisons to the selected method (frontal sinus trephination)
Author Response
Reviewer 1
The article addresses a clinically relevant issue (feline chronic rhinosinusitis - FCRS) with practical significance for veterinary clinicians.
Comments:
1) The rationale for prolonged glucocorticoid use, despite potential immunosuppressive side effects, was not discussed.
Response) Thank you for your thoughtful comment. We have added an explanation regarding the rationale and dosing strategy of glucocorticoid therapy in this case. Prednisolone was administered at an anti-inflammatory dose (0.5 mg/kg PO BID) rather than an immunosuppressive dose, and the dosage was tapered to 0.3 mg/kg PO BID by the first month. The medication was discontinued following the frontal sinus trephination procedure. This information has been added to the Case Description section (line 97). Postoperative treatment is already described in line 133.
Line 97-103
Medical therapy included marbofloxacin (2 mg/kg PO q24h), doxycycline (5 mg/kg PO q12h), azithromycin (5 mg/kg PO q24h, tapered to q72h after five days), bromhexine (1 mg/kg PO q12h), and omeprazole (1 mg/kg PO q12h). Prednisolone was administered at an anti-inflammatory dose (0.5 mg/kg PO q12h) and gradually tapered to 0.3 mg/kg PO q12h over approximately one month. Additionally, at-home nebulization with sterile saline was recommended once or twice daily.
2) Limited discussion on why repeated bacterial cultures were negative despite positive PCR findings.
Response) Thank you for raising this important point. As mentioned in line 172, it is well-documented that the pathogens identified in feline chronic rhinosinusitis (FCR) via PCR, culture, and histopathology often do not match, and the exact reason for these discrepancies remains unclear. However, possible explanations include the suppression of bacterial growth in culture media due to concurrent antibiotic administration or the involvement of fastidious organisms such as anaerobic bacteria. To address this, we have added a supplementary paragraph at line 202 in the Discussion section.
Line 202-208
In addition, due to the chronic nature of FCR, patients are often exposed to antibiotics or antifungal agents for prolonged periods, which may reduce the viability of pathogens in culture media. Moreover, the involvement of fastidious anaerobic bacteria cannot be excluded, as commonly used hospital transport systems and standard laboratory growth media may not provide optimal conditions for their proliferation [16]. In the present case, the inability to consistently isolate suspected pathogens through culture is presumed to be attributable to these factors.
3) Limited discussion of alternative surgical approaches or comparisons to the selected method (frontal sinus trephination)
Response) Thank you for your valuable comment. We have expanded the Discussion section to include a comparison with alternative surgical approaches previously reported in cats with FCR or FCRS. Specifically, we described outcomes from cases treated with frontal sinus obliteration using bone cement and radical turbinectomy, as well as cases managed with autogenous fat graft sinus ablation and ethmoid conchal curettage. These techniques, while potentially more definitive, are considerably more invasive and do not consistently lead to symptom resolution. A new paragraph has been added at line 230 to reflect this comparison.
Line 230-239
Seventeen cats diagnosed with FCR or FCRS have been treated using frontal sinus obliteration with bone cement and radical turbinectomy [24]. Among them, two cats showed marked clinical improvement, twelve exhibited a 10–70% reduction in clinical signs, and three showed minimal or no response despite surgical intervention. Another reported approach involved ethmoid conchal curettage combined with autogenous fat graft sinus ablation, in which two of six cases demonstrated clinical improvement [25]. Although these techniques may offer more definitive resolution compared to temporary frontal sinus trephination, they are significantly more invasive and may not result in consistent clinical benefits. Therefore, such procedures may be considered as alternative options in cases where symptoms persist or recur following initial sinus trephination and flushing.

Reviewer 2 Report
Comments and Suggestions for Authors
This case report offers a well-written and thoughtfully structured account of a feline patient with refractory chronic rhinosinusitis (FCRS) who was successfully treated through frontal sinus trephination and repeated irrigation. The subject is highly relevant, particularly given the persistent challenges in managing FCRS in cats and the relatively limited literature documenting long-term outcomes following surgical treatment.
The manuscript is concise, logically organized, and provides clear descriptions of the patient’s clinical presentation, diagnostic imaging, treatment strategy, and follow-up. The figures are well-chosen and enhance the narrative by supporting key details. The discussion effectively situates the case within the broader context of existing veterinary research and reflects a sound understanding of both the diagnostic complexities and therapeutic considerations involved in managing FCRS.
I have only a few minor editorial suggestions for the authors to consider:
- On page 2, line 62, there appears to be a stray period by itself, which should be deleted for consistency.
- On page 6, line 222, there is an extra closing parenthesis in the sentence:
“We obtained the consent of the owner and veterinarians) for medical reasons only.”
This parenthesis should be removed.
- On page 7, line 267, the DOI for Reference 14 ends with a double period. Please correct this to read: DOI: 10.1371/journal.pone.0180299.
Overall, this is a valuable and well-presented case report that contributes meaningful clinical insight. I recommend acceptance with minor revisions.
Author Response
Reviewer 2
This case report offers a well-written and thoughtfully structured account of a feline patient with refractory chronic rhinosinusitis (FCRS) who was successfully treated through frontal sinus trephination and repeated irrigation. The subject is highly relevant, particularly given the persistent challenges in managing FCRS in cats and the relatively limited literature documenting long-term outcomes following surgical treatment.
The manuscript is concise, logically organized, and provides clear descriptions of the patient’s clinical presentation, diagnostic imaging, treatment strategy, and follow-up. The figures are well-chosen and enhance the narrative by supporting key details. The discussion effectively situates the case within the broader context of existing veterinary research and reflects a sound understanding of both the diagnostic complexities and therapeutic considerations involved in managing FCRS.
I have only a few minor editorial suggestions for the authors to consider:
1) On page 2, line 62, there appears to be a stray period by itself, which should be deleted for consistency.
Response) Thank you for pointing this out. In accordance with your suggestion, the extraneous period on page 2, line 62 has been removed to improve formatting consistency.
2) On page 6, line 222, there is an extra closing parenthesis in the sentence:
“We obtained the consent of the owner and veterinarians) for medical reasons only.”
This parenthesis should be removed.
Response) Thank you for your careful review. As suggested, we have removed the extraneous closing parenthesis in the sentence on line 222 to correct the typographical error.
3) On page 7, line 267, the DOI for Reference 14 ends with a double period. Please correct this to read: DOI: 10.1371/journal.pone.0180299.
Response) We appreciate your attention to detail. The DOI for Reference 14 has been corrected to remove the extra period and now reads accurately as: DOI: 10.1371/journal.pone.0180299.
Overall, this is a valuable and well-presented case report that contributes meaningful clinical insight. I recommend acceptance with minor revisions.
Response) We sincerely thank the reviewer for the positive and encouraging feedback. We appreciate your recognition of the manuscript’s clinical relevance, structure, and clarity. Your constructive comments have been addressed carefully, and we believe they have improved the quality and readability of the manuscript.

Reviewer 3 Report
Comments and Suggestions for Authors
I thank the authors for this well-prepared and interesting manuscript describing the use of a surgical approach in the management of a cat with refractory chronic rhinitis. Below are several questions, suggestions, and concerns that I believe could improve the manuscript’s clarity and scientific rigor:
- L66: Please clarify how the diagnosis of FHV-1 was established. Was this based solely on clinical signs and history, or was it confirmed by virus-specific testing such as PCR or viral culture?
- L67: Did the cat receive all the mentioned treatments concurrently, or were they administered sequentially? A clearer treatment timeline would aid the reader’s understanding.
- L70: Despite the reported unremarkable findings, I suggest including the complete CBC and serum biochemistry results as supplementary material, along with the laboratory reference intervals.
- L80: Please elaborate on the technique used to collect the nasal discharge exudate for PCR testing. Specifically, how was sterile collection ensured while minimizing contamination from the oral flora? This is crucial for interpreting the PCR results.
- L82: A more detailed discussion of the clinical decision-making following the initial nasal flush would be helpful. In particular, the rationale for administering a combination of three antibiotics despite a negative bacterial culture result requires further justification.
- L86: Please specify the type of nebulizer used, the duration of each saline nebulization session, and how sterility of the saline was maintained in the home setting. Without proper hygiene, nebulization could risk introducing pathogens into the lower airways.
- L90: What is the proposed explanation for the newly developed atrophic changes in the right nasal turbinates within just 35 days? Given that the first CT already followed a chronic disease course and did not show such changes, this progression appears unusually rapid for typical chronic rhinitis.
- L124: Please provide details on the antibiotic used postoperatively, including the drug name and dosage.
- L148: Considering that FHV-1 can exhibit intermittent shedding and episodic clinical exacerbations, I question whether a lack of response to famciclovir alone is sufficient to rule out a contribution from FHV-1 to the clinical signs.
- L150: Relatedly, how was FHV-1 initially suspected as the primary etiology? Was there ever direct evidence of FHV-1 detection (e.g., PCR, culture), or was the diagnosis presumptive based on clinical presentation alone? If no confirmatory testing was performed and the response to antiviral treatment was poor, it is plausible that FHV-1 was never a contributing factor in this case.
- L170: I recommend citing the following reference:
Vangrinsven E, Duprez JN, Meex C, et al. Comparison of culture-dependent and -independent bacterial detection results on nasal swabs in dogs with nasal discharge. J Small Anim Pract. 2024;65(6):376–386. doi:10.1111/jsap.13734.
This study highlights the limitations of culture-dependent methods in nasal disease diagnostics and emphasizes that the presence or relative abundance of a microorganism on PCR does not necessarily indicate pathogenicity.
I would also suggest the author strengthen the discussion by mentioning that PCR, while sensitive, may detect normal flora or secondary colonizers rather than primary pathogens.
Author Response
Reviewer 3
I thank the authors for this well-prepared and interesting manuscript describing the use of a surgical approach in the management of a cat with refractory chronic rhinitis. Below are several questions, suggestions, and concerns that I believe could improve the manuscript’s clarity and scientific rigor:
1) L66: Please clarify how the diagnosis of FHV-1 was established. Was this based solely on clinical signs and history, or was it confirmed by virus-specific testing such as PCR or viral culture?
Response) Thank you for your comment. The diagnosis of FHV-1 in this case was based on the medical history provided by the owner. Although the owner was unable to specify the exact diagnostic methods used, they reported that the cat had been diagnosed with feline herpesvirus infection and received famciclovir treatment at a previous veterinary clinic. Unfortunately, we were unable to obtain direct access to the prior medical records or confirm whether PCR or viral culture had been performed. We have clarified this point in the manuscript accordingly.
Line 66-69
The diagnosis of feline herpesvirus-1 (FHV-1) was based on the medical history provided by the owner, who reported that the cat had been diagnosed with FHV-1 and treated with famciclovir at a previous veterinary clinic. However, the specific diagnostic method used (e.g., PCR or viral culture) could not be confirmed.
2) L67: Did the cat receive all the mentioned treatments concurrently, or were they administered sequentially? A clearer treatment timeline would aid the reader’s understanding.
Response) Thank you for pointing this out. The listed medications were not administered concurrently but rather represent various treatment attempts over the course of approximately one year. To avoid confusion, we have revised the relevant sentence to clarify that the drugs were used sequentially during prior treatment efforts. This clarification has been made in the Case Description section.
Line 70-73
Despite multiple treatment attempts over time, including the use of antibiotics, famciclovir, and glucocorticoids either separately or in combination, the cat continued to exhibit persistent mucopurulent nasal discharge and sneezing that were refractory to therapy.
3) L70: Despite the reported unremarkable findings, I suggest including the complete CBC and serum biochemistry results as supplementary material, along with the laboratory reference intervals.
Response) Thank you for your helpful suggestion. In response, we have added the complete CBC and serum biochemistry results, including reference intervals. This additional information provides greater transparency regarding the diagnostic workup.
We added Table 1.
Table 1. Hematological Results of the Cat
|
|
Result |
Reference range |
|
Hematocrit (%) |
48.7 |
30.3–52.3 |
|
White Blood Cells (k/μL) |
10.07 |
2.87–17.02 |
|
Neutrophils (k/μL) |
4.5 |
2.30–10.29 |
|
Lymphocytes (k/μL) |
4.82 |
0.92–6.88 |
|
Monocytes (k/μL) |
0.22 |
0.05–0.67 |
|
Eosinophils (k/μL) |
0.47 |
0.17–1.57 |
|
Platelet (k/μL) |
262 |
151–600 |
|
Glucose (mg/dL) |
104 |
74-159 |
|
BUN (mg/dL) |
21 |
16-36 |
|
Creatinine (mg/dL) |
1.3 |
0.8-2.4 |
|
ALT (U/L) |
67 |
12-130 |
|
ALP (U/L) |
28 |
14-111 |
|
GGT (U/L) |
0 |
0-4 |
|
Total protein (g/dL) |
8.2 |
5.7–8.9 |
|
Albumin (g/dL) |
3.3 |
2.2–4.0 |
|
Globulin (g/dL) |
4.9 |
2.8–5.1 |
|
Phosphorus (mg/dL) |
4.7 |
3.1-7.5 |
|
Calcium (mg/dL) |
10.1 |
7.8-11.3 |
BUN= Blood Urea Nitrogen, ALT= Alanine Aminotransferase, ALP= Alkaline Phosphatase, GGT= Gamma-Glutamyl Transferase
4) L80: Please elaborate on the technique used to collect the nasal discharge exudate for PCR testing. Specifically, how was sterile collection ensured while minimizing contamination from the oral flora? This is crucial for interpreting the PCR results.
Response) Thank you for this important question. At the time of sample collection, the patient was intubated with a tracheal tube under general anesthesia for CT imaging. The nasal area was first disinfected, and the patient’s head was positioned downward to facilitate drainage. A forced nasal flush was then performed by instilling sterile saline through both nostrils using a 10 mL syringe. The saline, along with the exudate, exited through both the nasal cavity and oral cavity. The purulent nasal discharge flowing from the nostrils was collected using a sterile swab and immediately placed in a PCR-specific transport medium. These steps were taken to reduce the risk of oral contamination and ensure sterile sample acquisition for molecular testing.
Line 90-95
Nasal flushing was performed while the patient was intubated with a tracheal tube. After disinfecting the nasal region, the patient was positioned in sternal recumbency with the head angled downward. Sterile saline was forcefully flushed into both nostrils using a 10 mL syringe. The resulting lavage fluid and exudate drained through the nasal and oral cavities. Purulent nasal discharge exiting from the nostrils was collected using a sterile swab and transferred into a PCR-specific transport medium.
(This image depicts the sampling process carried out in the present case.)
5) L82: A more detailed discussion of the clinical decision-making following the initial nasal flush would be helpful. In particular, the rationale for administering a combination of three antibiotics despite a negative bacterial culture result requires further justification.
Response) Thank you for your insightful comment. As also addressed in our response to Reviewer 1's second question, we did not interpret the negative culture results as definitive evidence for the absence of bacterial involvement. This is because PCR testing of the nasal exudate detected the presence of bacterial species, and culture results can be affected by prior antibiotic use or the presence of fastidious organisms. In light of the persistent clinical signs and the possibility of bacterial contribution, a combination antibiotic regimen was initiated to provide broad-spectrum coverage. In addition, mucosal damage from repeated flushing procedures raised concerns about secondary infection risk, further supporting the use of antibiotics. Prednisolone was administered at an anti-inflammatory dose to reduce mucosal inflammation, and bromhexine was added to decrease the viscosity of the nasal secretions and improve drainage.
Line 202-208
In addition, due to the chronic nature of FCR, patients are often exposed to antibiotics or antifungal agents for prolonged periods, which may reduce the viability of pathogens in culture media. Moreover, the involvement of fastidious anaerobic bacteria cannot be excluded, as commonly used hospital transport systems and standard laboratory growth media may not provide optimal conditions for their proliferation [16]. In the present case, the inability to consistently isolate suspected pathogens through culture is presumed to be attributable to these factors.
6) L86: Please specify the type of nebulizer used, the duration of each saline nebulization session, and how sterility of the saline was maintained in the home setting. Without proper hygiene, nebulization could risk introducing pathogens into the lower airways.
Response) Thank you for this important point. We recommended that the owner use a simple, household mesh-type nebulizer, which is commonly available and easy to operate. The owner was instructed to use sterile, single-use saline intended for injection or ophthalmic use for each session. After each use, the nebulizer was thoroughly rinsed with running water, completely dried in a clean, well-ventilated area with adequate sunlight, and then stored in a UV sterilizer designed for baby bottles. These hygiene measures were emphasized to minimize the risk of contamination and ensure the safe use of nebulization therapy in the home setting.
7) L90: What is the proposed explanation for the newly developed atrophic changes in the right nasal turbinates within just 35 days? Given that the first CT already followed a chronic disease course and did not show such changes, this progression appears unusually rapid for typical chronic rhinitis.
Response) Thank you for your thoughtful comment. Based on the available diagnostic results at the time, we were unable to definitively determine the cause of the rapid progression of turbinate atrophy. However, we also considered the unusually fast deterioration to be clinically significant and concerning. As a result, we included fungal rhinitis in our list of differential diagnoses and initiated empirical antifungal therapy to address the possibility of an undetected fungal component.
8) L124: Please provide details on the antibiotic used postoperatively, including the drug name and dosage.
Response) Thank you for your comment. The following antibiotics were administered postoperatively: clindamycin at 11 mg/kg PO q12h, doxycycline at 5 mg/kg PO q12h, and azithromycin at 7 mg/kg PO q72h. These agents were selected to provide broad-spectrum coverage based on the suspected pathogens identified by PCR and the chronicity of the infection.
Line 140-142
The cat was treated with terbinafine (30 mg/kg PO, q24h), fluconazole (1.25 mg/kg PO, q12h), clindamycin (11mg/kg PO q12h), doxycycline (5 mg/kg PO q12h), and azithromycin (7mg/kg PO q72h).
9) L148: Considering that FHV-1 can exhibit intermittent shedding and episodic clinical exacerbations, I question whether a lack of response to famciclovir alone is sufficient to rule out a contribution from FHV-1 to the clinical signs.
Response) Thank you for your thoughtful comment. The owner reported that famciclovir had been administered for an extended period prior to referral but had failed to produce noticeable clinical improvement. Furthermore, two separate PCR tests for feline herpesvirus-1 conducted at our hospital returned negative results. While we recognize the possibility of intermittent viral shedding, these consistent negative findings, along with the fact that the patient's condition improved significantly without any antiviral treatment during our management, suggest that FHV-1 was unlikely to have been a direct contributor to the ongoing clinical signs. We have added this explanation to the Discussion section to clarify our clinical reasoning.
Line 166-171
In the present case, the patient had a previous diagnosis of feline herpesvirus-1 (FHV-1), which was initially considered a potential underlying cause. However, the lack of clinical improvement despite long-term famciclovir administration, combined with two negative PCR results for FHV-1 and the marked clinical improvement achieved at our hospital without the use of antiviral therapy, suggest that FHV-1 was unlikely to have played a direct role in the ongoing clinical signs.
10) L150: Relatedly, how was FHV-1 initially suspected as the primary etiology? Was there ever direct evidence of FHV-1 detection (e.g., PCR, culture), or was the diagnosis presumptive based on clinical presentation alone? If no confirmatory testing was performed and the response to antiviral treatment was poor, it is plausible that FHV-1 was never a contributing factor in this case.
Response) Thank you for your observation. As noted in our response to Comment 1, the initial diagnosis of feline herpesvirus-1 (FHV-1) was based solely on the medical history provided by the owner. The owner was unable to specify the diagnostic method used but stated that the cat had been diagnosed with FHV-1 and treated with famciclovir at a previous veterinary clinic. Unfortunately, no direct medical records were available to confirm whether PCR or viral culture had been performed.
Line 66-69
The diagnosis of feline herpesvirus-1 (FHV-1) was based on the medical history provided by the owner, who reported that the cat had been diagnosed with FHV-1 and treated with famciclovir at a previous veterinary clinic. However, the specific diagnostic method used (e.g., PCR or viral culture) could not be confirmed.
11) L170: I recommend citing the following reference:
Response) Thank you for the helpful recommendation. We have incorporated the suggested reference into the Discussion section to support our comments regarding the limitations of culture-dependent methods in diagnosing nasal infections.
Line 188-208
In particular, although PCR testing is a highly sensitive diagnostic method for detecting pathogens, it is more likely to detect normal flora or secondarily colonizing bacteria rather than the primary pathogen in nasal samples. The presence or relative abundance of microorganisms identified via PCR does not necessarily indicate pathogenicity [18]. It is also important to note that the detection of microorganisms by PCR does not necessarily imply pathogenicity, as some organisms may represent normal flora or secondary colonizers. Therefore, PCR findings must be interpreted cautiously and in conjunction with clinical signs, imaging, and treatment response. Similarly, culture-based diagnostics may yield false-negative results due to prior antimicrobial therapy or fastidious growth requirements, underscoring the need for a multimodal diagnostic approach in FCR cases.
In this case, clinical symptoms worsened despite treatment and adjustment based on PCR results, suggesting that pathogen-targeted therapy alone may not be sufficient for clinical recovery in FCR.
In addition, due to the chronic nature of FCR, patients are often exposed to antibiotics or antifungal agents for prolonged periods, which may reduce the viability of pathogens in culture media. Moreover, the involvement of fastidious anaerobic bacteria cannot be excluded, as commonly used hospital transport systems and standard laboratory growth media may not provide optimal conditions for their proliferation [16]. In the present case, the inability to consistently isolate suspected pathogens through culture is presumed to be attributable to these factors.
- Vangrinsven E, Duprez JN, Meex C, et al. Comparison of culture-dependent and -independent bacterial detection results on nasal swabs in dogs with nasal discharge. J Small Anim Pract. 2024;65(6):376–386. doi:10.1111/jsap.13734.
12) This study highlights the limitations of culture-dependent methods in nasal disease diagnostics and emphasizes that the presence or relative abundance of a microorganism on PCR does not necessarily indicate pathogenicity.
I would also suggest the author strengthen the discussion by mentioning that PCR, while sensitive, may detect normal flora or secondary colonizers rather than primary pathogens.
Response) Thank you for these important observations. We fully agree that both PCR and culture-based diagnostics have limitations in identifying causative pathogens in FCR. PCR, while highly sensitive, may detect non-pathogenic organisms, such as normal flora or secondary colonizers, rather than primary pathogens. Conversely, culture-based methods may yield false-negative results due to prior antimicrobial use or the inability to grow fastidious organisms. We have revised the Discussion section to emphasize that microbial test results must be interpreted in the context of clinical presentation, imaging findings, and treatment response. A paragraph reflecting this integrated perspective has been added to strengthen the discussion.
Line 192-198
It is also important to note that the detection of microorganisms by PCR does not necessarily imply pathogenicity, as some organisms may represent normal flora or secondary colonizers. Therefore, PCR findings must be interpreted cautiously and in conjunction with clinical signs, imaging, and treatment response. Similarly, culture-based diagnostics may yield false-negative results due to prior antimicrobial therapy or fastidious growth requirements, underscoring the need for a multimodal diagnostic approach in FCR cases.

Round 2
Reviewer 3 Report
Comments and Suggestions for Authors
I thank the author for answering my previous concern and question.
I do believe the manuscript have been largely improved.